# 5-Azacytidine-Mediated Modulation of the Immune Microenvironment in Murine Acute Myeloid Leukemia

**DOI:** 10.3390/cancers15010118

**Published:** 2022-12-25

**Authors:** Nancy D. Ebelt, Edwin R. Manuel

**Affiliations:** Department of Immuno-Oncology, Beckman Research Institute of the City of Hope, Duarte, CA 91010, USA

**Keywords:** hematologic malignancies, acute myeloid leukemia, hypomethylating agents, 5-azacytidine, resistance, relapse

## Abstract

**Simple Summary:**

Leukemic growth results in selective silencing of gene transcripts involved in immune detection, differentiation, and apoptosis. Hypomethylating agents (HMAs), such as 5-azacytidine, have been shown to reprogram the genetic landscape in various cancers to restore both transgene and endogenous transcript expression. In this study, we show that 5-azacytidine treatment re-establishes both luciferase (transgene) and immune-related (endogenous) transcript expression in murine leukemia models, highlighting the broad applicability of HMAs in rescuing tumor suppressor and tumor antigen gene expression and enhancing cancer susceptibility to immunity. In addition, the immune microenvironment undergoes alterations after 5-azacytidine treatment, giving additional insight into the varied mechanisms behind 5-azacytidine-induced tumor growth control and possible combinatorial approaches to better control leukemia.

**Abstract:**

Cancer cells accumulate epigenetic modifications that allow escape from intrinsic and extrinsic surveillance mechanisms. In the case of acute myeloid leukemias (AML) and myelodysplastic syndromes, agents that disrupt chromatin structure, namely hypomethylating agents (HMAs), have shown tremendous promise as an alternate, milder treatment option for older, clinically non-fit patients. HMAs reprogram the epigenetic landscape in tumor cells through the reversal of DNA hypermethylation. Therapeutic effects resulting from these epigenetic changes are incredibly effective, sometimes resulting in complete remissions, but are frequently lost due to primary or acquired resistance. In this study, we describe syngeneic murine leukemias that are responsive to the HMA 5-azacytidine (5-Aza), as determined by augmented expression of a transduced luciferase reporter. We also found that 5-Aza treatment re-established immune-related transcript expression, suppressed leukemic burden and extended survival in leukemia-challenged mice. The effects of 5-Aza treatment were short-lived, and analysis of the immune microenvironment reveals possible mechanisms of resistance, such as simultaneous increase in immune checkpoint protein expression. This represents a model system that is highly responsive to HMAs and recapitulates major therapeutic outcomes observed in human leukemia (relapse) and may serve as a pre-clinical tool for studying acquired resistance and novel treatment combinations.

## 1. Introduction

Leukemia is a malignancy of the blood that is the most common cancer in children and teens, with the majority of cases diagnosed as acute lymphocytic leukemia (ALL) [1]. However, ≥90% of all leukemias are diagnosed in adults, with acute myeloid leukemia (AML) and chronic lymphocytic leukemia being among the most prominent [2]. For decades, the standard treatment for primary AML has been intensive chemotherapy with initial response rates reaching ≥80% for patients younger than 60 years of age [3]. However, long term survival rates remain low due to relapse, with a long-term survival rate of 40–50% for patients younger than 50 years of age, and less than 10% for patients older than 50 years of age [4]. In addition, many patients over 65 (which comprise the majority of AML cases diagnosed and AML deaths each year) are ineligible for chemotherapy or hematopoietic stem cell transplants due to infirmity [5].

An emergent hallmark of AML, and a growing number of other cancer types, is dysregulation of epigenetic gene silencing and activation [6]. AML generally displays low mutational burden, however, the majority of mutations occur in genes regulating epigenetic gene expression and processing, which contributes to the high heterogeneity of gene expression patterns between patients [7]. The use of hypomethylating agents (HMAs) to therapeutically re-express tumor suppressor and tumor antigen genes is currently front line therapy for AML patients that are un-fit for chemotherapy or hematopoietic stem cell transplant (HSCT) [8]. Overall response rates to the commonly used HMAs 5-Azacytidine (5-Aza) and decitabine (DAC) for myelodysplastic syndrome (MDS) and AML are around 50% and 15–20%, respectively [9,10]. The recent FDA approval of 5-Aza in combination with the BCl-2 inhibitor Venetoclax increases the response rate to 66% with a significant increase in overall survival, however approximately 50% of patients relapse within two years [11]. To combat pro-leukemic immune responses and to improve the longevity of HMA therapy, HMA combination with immune checkpoint antibodies is being investigated with some trials reporting modest improvements in response rate, but increased toxicity compared to HMA alone [12,13].

Clinical response of AML to HMA therapy has been correlated to a number of gene mutations and expression signatures [14], but the exact mechanisms of primary resistance or relapse have only begun to be understood. Furthermore, studies of the effects of HMA treatment on the immune microenvironment are incomplete with regard to many innate immune subsets and have reported conflicting results with regard to T-cell action, which may indicate both pro- and anti-tumor immune responses to HMA treatment [15]. These shortcomings highlight the need for leukemia models that recapitulate hypermethylation and sensitivity/resistance to HMAs to help illuminate the mechanisms of interplay between HMAs, leukemic cells, and the immune microenvironment. Pre-clinical studies in immune competent models can confirm clinical findings, delineate new biomarkers of response and resistance, and lead to hypothesis-driven therapeutic combinations for the treatment of human AML.

Transplantable mouse models of leukemia have many advantages: (1) the rapid onset of disease allows for immediate testing of experimental therapies, (2) immune competent mouse models support more mechanistic studies of disease progression (e.g., genetic manipulation, specific cell depletion) compared to correlative gene expression analysis in primary human samples, and (3) patient-derived xenograft or human cell line models using humanized mice lack fully functioning immune systems that may not correctly recapitulate leukemic interaction with the immune microenvironment [16].

While few transplantable mouse models of leukemia exist that recapitulate major aspects found in patients, due to forced transformation by mutagens and overexpression of oncogenes [17], the C1498 cell line arose from spontaneous AML. After engraftment via tail vein in C57Bl/6 mice, C1498 cells preferentially grow within the blood, bone marrow, and spleen [18,19]. The FBL3 cell line, induced by the Friend murine leukemia virus, presents as erythroleukemia, a rare form of AML [20,21]. Herein, we show that 5-Aza treatment rescued expression of an antigenic transgene (firefly luciferase, Fluc) in vitro and in vivo, and increased endogenous transcripts for antigen presentation machinery in both C1498 and FBL3 AML cell lines. Intravenously engrafted C1498-Fluc AML was sensitive to treatment with 5-azacytidine resulting in extended survival. Apoptosis of C1498-Fluc in vitro confirmed that 5-Aza has a direct effect on leukemic cell viability, possibly through re-expression of genes important for apoptosis. In the C57Bl/6 immune competent model, 5-Aza treatment increased CD4+ and CD8+ T-cells in the immune microenvironment as well as Mac3+MHCII+ macrophages. Frequencies of Ly6C+ and Ly6G+ monocytes and neutrophils decreased. Concomitantly, expression of PD-1, PD-L1, and CTLA-4 were differentially upregulated in these immune subsets possibly explaining why 5-Aza sensitivity is transient. Experiments in immunocompromised mice support that 5-Aza treatment also has a direct effect on C1498-Fluc growth that approximately 50% of mice respond to. The studies herein describe the sensitivity of the C1498 model to HMA treatment in vivo and are the first to delineate changes to the immune microenvironment caused by HMA treatment. The C1498 and FBL3 immune-competent, syngeneic mouse models are ideal for evaluating novel combinatorial approaches to determine their synergistic mechanisms and support their use in clinical trials.

## 2. Materials and Methods

### 2.1. Animals and Cell Lines

C57BL/6 mice (both male and female) were obtained from breeding colonies housed at the City of Hope (COH) Animal Research Center. NOD/SCID gamma (NSG) mice were obtained from Jackson Labs. Mice aged 7–8 weeks were used for all studies. For all studies, mice were handled according to standard institutional animal care and use committee (IACUC) guidelines (approved protocol #17128). The C1498 cell line was obtained from ATCC^®^ (TIB-49, Manassas, VA, USA). FBL3 cells were obtained from the DCTD tumor repository (NCI Frederick, MD, USA). Both cell lines were maintained in RPMI media supplemented with 10% FBS, 2mM L-glutamine and 100 units/mL penicillin, and 100 µg/mL streptomycin. Cell lines were passaged minimally (≤5 times) before implantation in mice.

### 2.2. Generation of Firefly Luciferase-Expressing Leukemia Cell Lines C1498-FLuc and FBL3-FLuc

Both C1498 and FBL3 leukemia cell lines were transduced with lentivirus carrying the firefly luciferase (FLuc) transgene with puromycin resistance (Kerafast, FCT006; Boston, MA, USA). Forty-eight hours following transduction, cells were selected in RPMI media containing 5 µg/mL puromycin (ThermoFisher; Waltham, MA, USA) for 1–2 weeks. Bulk resistant cells were expanded in puromycin-containing RPMI and confirmed to be bioluminescent following addition of 200 µg/mL D-luciferin (GoldBio; St. Louis, MO, USA). Bioluminescence was detected using an iBright imaging system (Thermofisher; Waltham, MA, USA) or a Lago X live imager (Spectral Imaging; Tucson, AZ, USA).

### 2.3. Mouse Challenge and Intravital Imaging

For intravital imaging studies, C1498-FLuc cells (7.5 × 10^4^) were introduced into mice through intraperitoneal (i.p.) injection. Two weeks following leukemic challenge, mice were administered 150 mg/kg D-luciferin (i.p.) 5 min before imaging with a Lago X live imaging system. Bioluminescent signal was quantified using Aura software (Spectral Imaging; Tucson, AZ, USA).

### 2.4. In Vitro 5-Aza Treatment Studies of Luciferase Activity and Viability in Leukemia Cell Lines

FLuc-transduced leukemia cell lines were seeded in 96-well flat-bottom plates at 1 × 10^5^ cells and then treated with 5-Aza or DMSO (diluent) at increasing concentrations in the range of 0 to 10 μM in complete RPMI media. At 24- and 48-h post-treatment, viability of treated cells was determined by trypan blue staining. Additionally, D-luciferin was added to replicate wells at a final concentration of 20 µg/mL and bioluminescence was detected using an iBright imaging system and quantified (densitometry) using ImageJ software.

### 2.5. In Vivo 5-Aza Treatment Studies and Survival Studies following Leukemia Challenge

For therapeutic studies, mice were implanted with 8 × 10^4^ C1498 cells via intravenous (i.v.) tail injection. Three days post-implant, mice were treated with 5 mg/kg 5-Aza or DMSO (in diluent) for 3 consecutive days by i.p. injection in a total volume of 500µL in 1× HBSS. For survival analysis, mice showing signs of disease progression following leukemic challenge (lethargy, paralysis, hunched, etc. or ovarian mass (females)) were immediately euthanized. Survival data is displayed as Kaplan–Meier curves.

### 2.6. Quantitative PCR Analysis of Antigen Presentation-Related, Endogenous Transcripts

One million human and murine leukemia cell lines were treated in vitro with 5 μM or 1 µM 5-Aza for 48 h in complete RPMI media, respectively. Total RNA from approximately 5 × 10^5^ cells was isolated using the Omega Biotek E. Z. N. A. Total RNA kit I (Norcross, GA, USA) with on column DNase I digestion of remaining genomic DNA. Total RNA was converted to cDNA using the Applied Biosystems High Capacity cDNA Reverse Transcription Kit (using random primers) (Applied Biosystems; Waltham, MA, USA). cDNA was amplified and fluorescently labeled using the Bioland Scientific 2X qPCR Master Mix (Low Rox) (Paramount, CA, USA) containing SYBR Green in the QuantStudio3 Real Time PCR System (Thermofisher; Waltham, MA, USA). Experiments were run with “no template” controls to control for nucleic acid contamination in water or primers, no data was used if sample CTs fell within five CTs of the “no template” controls. Primer sequences were published previously [22].

### 2.7. Flow Cytometric Analysis

For cells in culture: Two million C1498-Fluc cells were treated with 10 µM 5-Aza or vehicle for 48 h in culture. Live cells were counted, washed, and stained as follows.

In vivo cell isolation: Blood was collected by retro-orbital bleed (~200 µL) followed by RBC lysis with 1× Pharmlyse (BD Biosciences, Franklin Lakes, NJ, USA) for 3 min at room temperature. Freshly isolated spleens were mashed carefully against 70µm filters and washed with complete RPMI media to obtain splenocytes. Bone marrow was isolated from the femur by washing the intra-femoral space with complete RPMI media. Liver and lung cells were isolated after mechanical digestion with scalpels (#21) followed by shaking (200 rpm) in dissociation buffer (1 mg/mL collagenase IV (Sigma C5138), 0.25 mg/mL hyaluronidase (Sigma H3506), and 1% BSA in 1× HBSS w/o Ca^2+^ and Mg^2+^) for 30 min at 37 °C to produce single cell suspensions. RBC lysis with Pharmlyse followed isolation of single cells.

General: One million live cells were counted using trypan blue and first stained with a fixable viability dye (eBiosciences 65-0866-14, Thermo Fisher Scientific; Waltham, MA, USA) for 30 min at 4 °C. Cells were washed in flow wash buffer (PBS with 0.05% sodium azide and 3% FBS) and stained with surface antibodies for 40 min at 4 °C. Cells were fixed and permeabilized for intracellular staining using the BD Cytofix/Cytoperm™ kit (554714, BD Biosciences; San Jose, CA, USA). After final washes cells were filtered through 40-μm mesh strainer tubes (BD Biosciences; San Jose, CA, USA). Flow cytometry was performed on the BD Fortessa X20 cytometer, and the data were analyzed using FlowJo Version 10 (Becton, Dickinson & Co.; Franklin Lakes, NJ, USA). Flow cytometry antibodies used from BD Biosciences (San Jose, CA, USA) include: CD8-APC-R700 (564983), CD4-APC-H7 (5560181), Ly6G-BV605 (563005), Ly6C-FITC (561085), CD11b-APC (561690), PD-1-BV421 (562584), MHCII-PerCPCy5.5 (562363), Mac3-BUV395 (743186), CD11c-BV786 (563735), and CTLA4-PE (553720). Recombinant Anti-Firefly Luciferase antibody (ab185924) (Abcam; Waltham, MA, USA). PD-L1-SuperBright 645 (64-5982-82) and Goat-anti rabbit IgG-Alexa Fluor 647 (A32733) (Thermo Fisher Scientific; Waltham, MA, USA). Approximately one hundred thousand cells were analyzed per sample.

### 2.8. Annexin V staining for Apoptosis

One million C1498-FLuc cells in vitro were treated with vehicle (DMSO) or 5-Aza (5 μM) for 48 h. One hundred thousand live cells were counted, washed, and stained with anti-Annexin V antibody (Alexa Fluor 488) and propidium iodide (kit V13241) (Thermo Fisher Scientific; Waltham, MA, USA) and subjected to flow cytometric analysis on the BD Fortessa X20 cytometer. Data were analyzed using FlowJo Version 10 (Becton, Dickinson & Co.; Franklin Lakes, NJ, USA).

### 2.9. Splenocyte Co-Culture with MTT Assay for Viability

Two hundred and fifty thousand C1498-FLuc or FBL3-FLuc in culture were treated with vehicle or 5 μM 5-Aza for 24 h in complete RPMI media. Media including treatment was washed out and splenocytes from non-leukemic mice treated with vehicle or 5 mg/kg 5-Aza were cocultured with cell lines in full media. After 24 h of co-culture, MTT reagent (Thiazolyl Blue, Selleck Chemicals, Houston, TX, USA) was added to a final concentration of 1 mg/mL for 4 h to allow formation of formazan crystals. After incubation, media plus MTT was carefully removed and formazan crystals were dissolved in 400 μL of DMSO. Signal was read on a Biotek plate reader (Santa Clara, CA, USA) at 590 nm, and cellular background (630 nm) was subtracted from 590 nm readings for each well. Signal from splenocytes cultured alone was subtracted from wells with cell lines and splenocyte co-culture.

### 2.10. Statistics

All statistical analyses were performed using the Prism software by GraphPad (V9) (San Deigo, CA, USA). Unless otherwise indicated, all error bars represent standard error of the mean.

## 3. Results

### 3.1. FLuc Transgene Expression Is Silenced in FLuc-Transduced Murine AML Cell Lines

Introduction of transgenes can increase the immunogenic potential of leukemic cells, similar to the expression of certain germline or neoantigen proteins in spontaneously occurring cancers. For our studies, we have introduced the FLuc gene into two murine leukemia lines, C1498 and FBL3, via lentiviral transduction in order to gauge leukemic burden in vivo, but we first sought to assay the stability of transgene expression as insertion mosaicism and increased immunogenicity of the cell line can lead to clonal selection against expression through epigenetic silencing [23,24,25,26,27]. After antibiotic selection, FLuc expression was shown to be stable in both cell lines in vitro (Figure 1A). However, following implantation in mice, FLuc signal became highly variable as the AML progressed, with a majority of mice losing bioluminescent signal (Figure 1B). Despite mice completely losing signal, they still succumbed to leukemic disease (Figure 1C), suggesting that C1498 is still present, but has lost transgene expression. Interestingly, despite some mice retaining FLuc expression, survival curves did not differ between mice implanted with parental C1498 and C1498-FLuc. As Fluc expression has been shown to elicit an immune response and alter survival compared to parental cell lines in other models [23,24,25,26], it is possible that C1498-Fluc cells retaining high expression may have undergone selection for decreased MHCI-antigen presentation, a phenomenon observed in many human cancers [28]. We next sought to determine if HMA therapy could rescue (reprogram) transgene expression and possibly induce therapeutic effects as observed in patients.

### 3.2. 5-Aza Augments Bioluminescence of Ex vivo FLuc-Transduced AML Cell Lines

Aberrant methylation in leukemia cells affects endogenous gene expression in AML [29,30,31], and has also been observed with regard to transgene expression in C1498 [32]. Despite some mice retaining bioluminescent signal in initial experiments where mice were challenged intraperitoneally (i.p.), cells taken ex vivo and cultured from the blood of mice challenged by tail vein generally showed a complete loss of bioluminescent signal. We determined that culturing leukemia cells with 5-Aza ex vivo rescued bioluminescence (Figure 2A), however, higher concentrations of 5-Aza became cytotoxic (Figure 2B) resulting in lower luminescence due to cell death. Treatment of C1498-FLuc challenged mice with 5-Aza also rescued bioluminescence in vivo (*p* = 0.0172, Welch’s *t*-test) (Figure 2C). These data confirm that FLuc loss occurs through epigenetic silencing and suggests that rescue of antigen expression in human AML may explain in part, the therapeutic efficacy of HMA treatment in patients.

### 3.3. 5-Aza Induces Changes in Antigen Presentation-Related Endogenous Gene Expression

In addition to rescuing tumor antigen expression, HMA treatment may also reprogram expression of genes encoding machinery for antigen presentation [22]. Indeed, although some tumor types select for mutations in MHC I genes during immunological escape, MHC I gene mutations are uncommon in AML. One study found that there was no specific mutational signature involved with AML relapse after HSCT, although antigen presentation genes were downregulated at the transcript level [33]. Thus, mutational burden in AML remains low even after treatment, and HMA therapy may be able to restore antigen presentation in AML similar to what has been found in other cancers [22,34,35]. To determine the effects of 5-Aza treatment on antigen presentation, syngeneic mouse (C1498-Fluc and FBL3-Fluc) and human leukemia (HL-60 (acute promyelocytic leukemia, an AML subtype) and Jurkat (acute T-cell leukemia)) cell lines were treated with 5-Aza in vitro, and mRNA transcripts for antigen presentation genes were measured by quantitative PCR. Both C1498-FLuc and FBL3-FLuc cell lines showed significant increases in Tap1 and Lmp2 transcripts after treatment in vitro with 5-Aza, indicating increased potential to process and transport peptides for MHC I-dependent presentation (Figure 3A,B). In addition, FBL3-FLuc cells showed significant upregulation of MHC I itself (Figure 3B). HL-60 and Jurkat cell lines showed significantly increased expression of several genes important for antigen presentation including HLA-A, HLA-C, huβ2m, huTap1 and huLmp2 (Figure 3C,D). These results indicate epigenetic silencing of multiple genes involved in antigen presentation in both mouse and human leukemias. Although HL-60 and Jurkat cells represent leukemias other than AML, these data support aberrant epigenetic gene regulation as a general hallmark of leukemic transformation and progression [36,37,38]. To determine whether 5-Aza treatment can induce functional immunity against C1498-FLuc and FBL3-FLuc cell lines, cells in vitro were co-cultured with splenocytes harvested from non-leukemic mice treated with vehicle or 5 mg/kg 5-Aza. For C1498-FLuc, co-culture induced significant cell killing only in 5-Aza treated cells co-cultured with splenocytes from 5-Aza treated mice indicating increased immunity against C1498-FLuc only after 5-Aza treatment of both leukemic cells and immune cells (Figure 3E). For FBL3-FLuc, DMSO-treated cell lines induced significant killing from DMSO-treated splenocytes but not 5-Aza treated splenocytes, and 5-Aza treated cells induced no killing, indicating that 5-Aza treatment of cell lines or immune cells reduces FBL3-FLuc-specific immunity (Figure 3F). These data highlight differential effects of 5-Aza treatment on specific leukemias.

### 3.4. 5-Aza Treatment Extends Survival in Leukemic Mice

5-Aza treatment of AML in humans typically results in a 50% response rate with approximately half of responders experiencing relapse after complete remission [9,10,11]. It is imperative that the interplay between 5-Aza’s effects on AML cells and the immune microenvironment be deciphered in order to develop hypothesis-driven combinatorial strategies to overcome resistance. To evaluate the changes in immunity induced by 5-Aza, we first initiated therapeutic studies. Immune-competent C57Bl/6 mice were implanted with C1498-FLuc cells by i.v. tail vein route. On day 4 post-challenge, mice were given three consecutive treatments of 5-Aza (1 mg/kg or 5 mg/kg) or vehicle (0 mg/kg) and survival of the three groups was recorded (Figure 4A). Blood smears from mice at Day 17 post-challenge were used to confirm engraftment of C1498-Fluc as seen by blast-like cells that are easily differentiated from normal WBC (Figure 4B, Appendix A). 5-Aza treatment with 1 mg/kg did not significantly improve survival over DMSO vehicle (median survival, 25 vs. 23 days), however treatment with 5 mg/kg 5-Aza resulted in a significant improvement in survival compared to vehicle (median survival 29 vs. 23 days, *p* = 0.0042 Log-rank (Mantel-Cox) test, *p* = 0.0062 Gehan-Breslow-Wilcoxon test). The 5 mg/kg treatment is significantly higher than the dose found to rescue FLuc expression in vivo with i.p. challenge (5 μg/kg, Figure 2C), implying that more than rescue of Fluc expression, and possibly antigen presentation machinery expression, is needed to extend survival. In vitro treatment of C1498-FLuc cells with 5-Aza causes significantly increased apoptosis as evidenced by Annexin V positivity (Figure 4C), and in vivo we found that the prevalence of C1498-Fluc cells is significantly decreased in various compartments, including the blood (*p* = 0.0052, Welch’s *t*-test), spleen (*p* = 0.0056, Welch’s *t*-test), and lung (*p* = 0.0443, Welch’s *t*-test), indicating that extended survival is due to a decrease in overall leukemic burden (Figure 4D).

### 3.5. 5-Aza Extends Survival for a Subset of Leukemic, Immune-Deficient Mice

To determine whether 5-Aza extends survival through mostly immunological anti-leukemic effects or direct toxicity of C1498-FLuc cells, immune-deficient NOD scid gamma (NSG) mice were challenged with C1498-FLuc and treated with 5-Aza (5 mg/kg) in the same manner as the C57Bl/6 model. Tumor-burden at euthanization was confirmed through the presence of ovarian enlargement due to C1498-FLuc infiltration [19]. In NSG mice, 5-Aza treatment did not significantly extend survival (Figure 5A) (*p* = 0.1531 Log Rank (Mantel-Cox) test, *p* = 0.1374 Gehan-Breslow-Wilcoxon test) indicating that prolonged survival after 5-Aza treatment in the C57Bl/6 immunocompetent model is possibly due to anti-leukemic effects involving a functional immune system. However, 50% of 5-Aza treated mice showed markedly increased survival (median survival 26 vs. 46.5 days for 0 mg/kg vs. 5 mg/kg treated mice, respectively), indicating that there is direct toxicity of 5-Aza on C1498-Fluc cells in these responsive mice. Interestingly, when comparing survival curves between NSG and C57Bl/6 models, survival for vehicle treated mice in NSG mice was significantly improved compared to vehicle-treated mice in the immunocompetent C57Bl/6 model (Figure 5B) (*p* = 0.0246 Log Rank (Mantel-Cox) test, *p* = 0.0339 Gehan-Breslow-Wilcoxon test). The difference in median survival, however, was quite small and may be due to niche differences between the two mouse models beyond immune competency. Comparison of 5-Aza treatment between C57Bl/6 and NSG mice shows that 50% of mice show extended survival in NSG immunocompromised mice beyond that seen in the C57Bl/6 mice (median survival 29 vs. 46.5 days 5 mg/kg treated C57Bl/6 and NSG mice, respectively) (Figure 5C), although statistical significance was again not reached (*p* = 0.0596 Log Rank (Mantel-Cox) test, *p* = 0.1306 Gehan-Breslow-Wilcoxon test). Based on these data, we speculate that increased survival of C1498-FLuc engrafted C57Bl/6 mice after 5-Aza treatment involves both direct toxicity and immune-related effects on leukemic cells. In the next section we explore 5-Aza effects on the immune system to determine possible mechanisms of action.

### 3.6. 5-Aza Treatment in Leukemic Mice Alters Immune Subset Frequencies and Checkpoint Protein Expression

Previous clinical studies in leukemia have shown decreased methylation and increased expression of immune checkpoint molecules such as programmed death receptor 1 (PD-1), PD ligand 1 (PD-L1), and Cytotoxic T-lymphocyte Antigen-4 (CTLA-4) in patients after HMA treatment, prompting combination therapy of HMAs with immune checkpoint blockade antibodies [39,40]. It has not been determined mechanistically whether expression of checkpoint molecules is the cause of HMA resistance or relapse. Consistent with previous studies, we confirmed high expression of PD-L1 [41] as well as some expression of PD-1 and CTLA-4 on untreated C1498-FLuc cells grown in vitro (Figure 6A). This expression was not significantly changed by in vitro 5-Aza treatment. In C57Bl/6 mice, however, C1498-FLuc expression of these proteins varied and was dependent on the tissue in which the C1498-Fluc cells were sampled. Figure 6B shows that in vivo, PD-1 was the most highly expressed immune checkpoint protein in all tissues sampled except for bone marrow, and treatment with 5-Aza significantly increased PD-1 expression on C1498-FLuc in blood (Welch’s *t*-test, *p* = 0.0122). Expression of PD-L1 on C1498-Fluc was much lower in all tissues (compared to in vitro levels) except for the spleen where it was very highly expressed. 5-Aza treatment was not shown to alter PD-L1 expression levels on C1498-FLuc in any of the sampled tissues. Finally, CTLA-4 expression on C1498-FLuc was low in all tissues; however, 5-Aza treatment further decreased CTLA-4 levels significantly in C1498-FLuc found in blood, spleen, and lung (Welch’s *t*-test, *p* = 0.0074, *p* = 0.0022, and *p* = 0.0167) (Figure 6B). While the significance of PD-1 expression on AML cells is unclear, these data suggest that PD-L1/PD-1 interaction between C1498-Fluc cells and nearby T-cells may play a role in AML progression in the spleen.

Analysis of cells in the blood of leukemic mice after 5-Aza treatment shows significant increases in both CD4+ and CD8+ T-cells (Welch’s *t*-test, *p* = 0.0102, *p* = 0.0086, respectively) (Figure 6C). Within these subsets, there were also modest but significant increases in percentages of CD4+ T-cells that are PD-1 and PD-L1 positive (Welch’s *t*-test, *p* = 0.0101 and *p* = 0.0294), and CD8+ T-cells that are PD-L1 positive (Welch’s *t*-test, *p* = 0.0247) (Figure 6D). Frequencies of CD4+ and CD8+ T-cells in the spleen were unchanged after 5-Aza treatment (Appendix A), but PD-1 expression on CD8+ T cells (Welch’s *t*-test, *p* = 0.0187) as well as PD-L1 expression on both CD4+ (Welch’s *t*-test, *p* = 0.0265) and CD8+ (Welch’s *t*-test, *p* = 0.0102) T-cells were significantly increased after 5-Aza treatment (Appendix A). These data are consistent with clinical data from primary AML samples and suggest that 5-Aza treatment may increase activation of T-cells initially, but simultaneous hypomethylation of PD-1 and PD-L1 could lead to dampened immune responses shortly after. These data are also consistent with clinical findings that HMAs increase CD8+ T-cell function [42], but are eventually met with acquired resistance, and might mechanistically explain transient 5-Aza-induced tumor control.

5-Aza treatment in leukemic mice also affected frequency and expression of immune checkpoint molecules on innate immune cells. In the blood of leukemic mice Ly6C+ monocytes and Ly6G+ neutrophils were significantly decreased (Welch’s *t*-test, *p* = 0.0006 and *p* = 0.0243, respectively) after 5-Aza treatment while Mac3+MHCII+ macrophages were significantly increased (Welch’s *t*-test, *p* < 0.0001) (Figure 6E). Neutrophils showed a significant increase in PD-1 expression (Welch’s *t*-test, *p* = 0.0048), monocytes and neutrophils showed significant increases in PD-L1 expression (Welch’s *t*-test, *p* = 0.0214 and *p* = 0.0048), and neutrophils showed increased expression of CTLA-4 while macrophages showed decreased expression of CTLA-4 (Welch’s *t*-test, *p* = 0.0003 and *p* = 0.0033) (Figure 6F). In spleen, Ly6G+ neutrophils were significantly decreased after 5-Aza treatment (Welch’s *t*-test, *p* = 0.0009), but other innate immune cell frequencies remained unchanged (Appendix A). However, in Ly6C+ monocytes, PD-1 expression was significantly decreased while PD-L1 expression was significantly increased (Welch’s *t*-test, *p* = 0.0003 and 0.0017). On Ly6G+ neutrophils, PD-1 expression was significantly decreased while both PD-L1 and CTLA-4 expression were significantly increased (Welch’s *t*-test, *p* = 0.0233, *p* = 0.0013, and *p* = 0.0161, respectively). Finally, Mac3+MHCII+ macrophages showed a modest but significant increase in PD-1 expression after 5-Aza treatment (Welch’s *t*-test, *p* = 0.0461) (Appendix A). As myeloid derived suppressor cells (MDSC) are expanded in AML and MDS [43,44], a reduction in Ly6C+/Ly6G+ cells in this model may indicate a reduction in MDSCs after 5-Aza treatment, although, as with the CD4+ and CD8+ T-cells, increased expression of PD-L1 and CTLA-4 on MDSC may intensify their suppressive function leading to eventual resistance.

## 4. Discussion

Previous studies involving the in vitro passage of the murine leukemia cell line C1498 observed the loss of antigenic transgene expression through hypermethylation of either the promoter or transgene itself. This was supported by experiments showing that HMA treatment could completely rescue transgene expression [32]. However, additional experiments to evaluate the effects of HMA treatment on transgene expression and leukemic burden in vivo were not performed. A more recent study describes sensitivity of C1498 in vivo to guadecitabine as well as combination of guadecitabine with a DC/AML fusion vaccine. However, these studies employed retro-orbital implantation of C1498 wherein the majority of tumor growth is within the eye and not in typical AML target tissues [45]. We are the first to show that 5-Aza treatment rescues C1498-FLuc transgene expression in vivo, as well as decreases leukemic burden and increases survival in C57Bl/6 mice with tail vein engrafted C1498-FLuc. C1498-Fluc are found to infiltrate blood, spleen, bone marrow and lungs in this model, reminiscent of human AML [46,47]. The FBL3-Fluc model responds similarly to 5-Aza in vitro, but will require extensive optimization before utilization for in vivo studies as its hyper-aggressive growth prevents therapeutic intervention [32].

Within this study we found that the C1498-Fluc transplantable, immune-competent model replicates many hallmarks of the response of human AML to 5-Aza treatment. First, a single round of 5-Aza (three consecutive days) resulted in a significant increase in survival, but all mice eventually succumbed. As HMA treatment of human AML involves an extended period of therapy [48], it is possible that the C1498-Fluc model may reflect further extension in survival if treatment is prolonged.

Despite a short treatment regimen, however, important insights were gained into the mechanisms of 5-Aza-induced efficacy using the C1498-FLuc model. 5-Aza treatment of NSG mice revealed that 50% of mice responded to 5-Aza with extended survival, indicating direct, but not fully penetrant effects of 5-Aza on C1498-Fluc cell viability. Analysis of the immune microenvironment in the blood and spleen of C57Bl/6 mice engrafted with C1498-Fluc and treated with 5-Aza supports a role for the immune system as well. Frequencies of CD4+ and CD8+ T-cells were increased in the blood of 5-Aza-treated leukemic mice compared to vehicle, but with concomitant increases in the expression of PD-1 on CD4+ T cells in blood and CD8+ T-cells in spleen, and PD-L1 on both CD4+ and CD8+ T-cells in blood and spleen, consistent with clinical data from HMA treated patients [39,42]. Ly6C+ and Ly6G+ cells were significantly decreased in blood (Ly6G+ only in spleen) also with increases in expression of PD-L1 and CTLA-4 in blood and spleen, and with increased PD-1 expression in blood but decreased expression in spleen. The probability that these subsets are MDSC, and not functional monocytes or neutrophils, is supported by their inverse correlation to T-cell frequencies [49]. Thus, 5-Aza-induced reduction in MDSC frequency is a likely mechanism for increased T-cell frequency. However, the caveat of increased suppressive function (as evidenced by immune checkpoint protein expression), emphasizes the need for combination therapy and is a probable cause of acquired resistance to 5-Aza [40]. While more mechanistic studies of immune microenvironment contribution to C1498 disease progression will be needed to make any definitive conclusions, such as depletion of these subsets in our model, these data together indicate anti-leukemic effects by certain immune subsets after 5-Aza treatment, as well as direct toxicity of 5-Aza on C1498 cells. Confirmation of these data in primary AML samples and humanized PDX mouse models will be helpful to fully dissect the mechanisms of interplay between leukemic cells and the immune microenvironment.

Clinical trials combining HMA with therapeutic immune checkpoint antibodies are actively underway. As a first-line therapy, the combination of 5-Aza with durvalumab (anti-PD-L1) did not improve response, survival, or durability compared to 5-Aza alone [12]. In relapsed AML patients however, the combination of 5-Aza with nivolumab (anti-PD-1) and ipilimumab (anti-CTLA-4) significantly increased median overall survival compared to 5-Aza alone [13]. Data from our studies support that blocking PD-L1/PD-1 signaling concurrent with CTLA-4 signaling may be necessary to overcome resistance to HMA as these proteins were simultaneously upregulated after 5-Aza treatment.

Despite these encouraging results, the 1-year overall response rate remains low (45%) with HMA and immune checkpoint blockade combination therapy, and improvement will hinge upon hypothesis driven experimentation. Data from our NSG experiments imply that therapeutics to overcome internal AML resistance signaling may improve HMA and immune checkpoint blockade combination, demonstrating that this pre-clinical model of HMA-sensitive AML can be an invaluable tool to further investigate the mechanisms and possible remedies for HMA resistance. We anticipate confirmation of these findings in primary human leukemia cells and in humanized patient-derived xenograft mice in future experiments, and hope these combined data will propel combinatorial strategies for clinical trial.

## 5. Conclusions

In this study, we have evaluated the effects of HMA treatment on leukemia cells and observed significant reprogramming of both transgene and endogenous gene expression. Interestingly, we found that while 5-Aza treatment enhances the immunogenicity of leukemia cells and upregulates antigen presentation-related transcripts, we also observed simultaneous upregulation of immune checkpoint proteins on the surface of both adaptive and innate immune subsets. We conclude that 5-Aza treatment may act as a double-edged sword, providing anti-leukemic effects while also inducing immune-suppressive pathways to dampen efficacy and increase resistance. For future work, our HMA treatment model will be instrumental in evaluating novel combinatorial approaches using HMAs. Our results strongly suggest simultaneous treatment with immune checkpoint blockade therapy, which may provide additional insight and support for current clinical trials evaluating this specific combination.

## Figures and Tables

**Figure 1 cancers-15-00118-f001:**
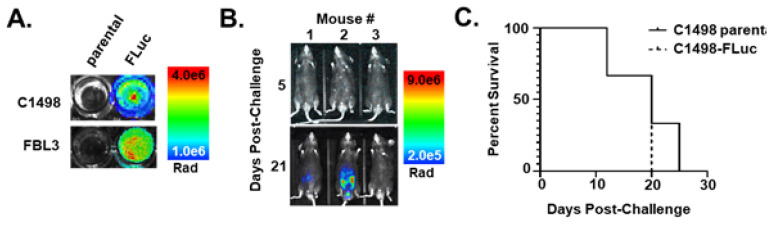
Introduction of FLuc-transduced leukemia cell lines in vivo silences the FLuc transgene. (**A**) Pseudocolor representations of AML syngeneic cell lines transduced with lentivirus containing FLuc after addition of D-Luciferin (200 µg/mL) to cells in culture (Rad (radiance) = photons/s/cm^2^/sr). (**B**) Pseudocolor representations of intravital luminescence imaging of mice (*n* = 3) challenged intraperitoneally (i.p.) with C1498-FLuc cells after i.p. injection of D-Luciferin (150 mg/kg). Days 5 and 21 post-challenge are shown. (**C**) Kaplan–Meier survival curve of mice challenged intravenously (i.v.) via tail vein with parental C1498 or C1498-FLuc cells. Within each group mice were challenged with one of three different cell numbers: 5 × 10^4^, 2.5 × 10^5^, or 1 × 10^6^ cells/mouse.

**Figure 2 cancers-15-00118-f002:**
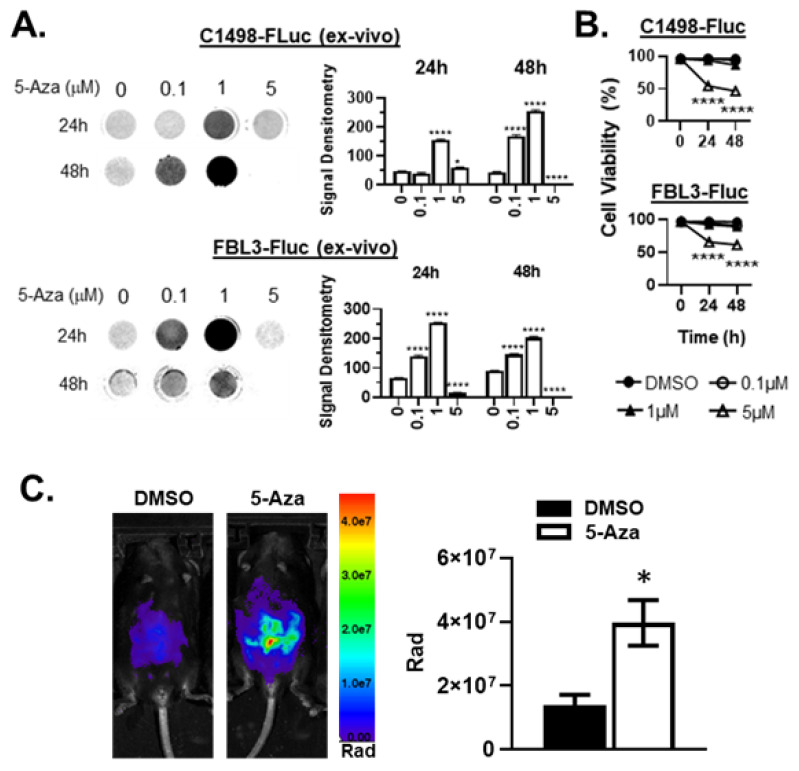
5-Aza treatment rescues bioluminescence ex vivo and in vivo in AML-FLuc cell lines. (**A**) C57Bl/6 mice were implanted i.v. with C1498-FLuc or FBL3-FLuc cell lines and AML was allowed to develop (C1498-FLuc: 8 × 10^4^ cells for 18 days, FBL3-FLuc: 5 × 10^4^ cells for 6 days). Blood was taken from challenged mice and grown in culture after RBC lysis until primary cells were eliminated. Cells were then incubated in increasing concentrations of 5-Aza for 24 and 48 h. D-luciferin was added at a concentration of 150µg/mL and luminescence was read. Bar graphs represent signal intensity as densitometry, *n* = 3 biological replicates/group. 2-way ANOVA followed by Tukey’s test for multiple comparisons, all statistics shown compared to group “0”, * *p* < 0.05, **** *p* < 0.0001. (**B**) Ex vivo cell viability was assayed by trypan-blue exclusion after incubation in increasing concentrations of 5-Aza for 24 and 48 h. 2-way ANOVA followed by Tukey’s test for multiple comparisons, all statistics shown compared to group “0”, **** *p* < 0.0001. (**C**) C57Bl/6 mice were challenged i.p. with 6 × 10^5^ C1498-FLuc cells. After 12 days, three consecutive treatments of Aza (5 µg/kg) were given i.p. and intravital luminescence imaging was performed 24 h after the final treatment. Pseudocolor images are shown of representative mice, *n* = 4 mice per group. *p* = 0.0172, unpaired *t*-test with Welch’s correction.

**Figure 3 cancers-15-00118-f003:**
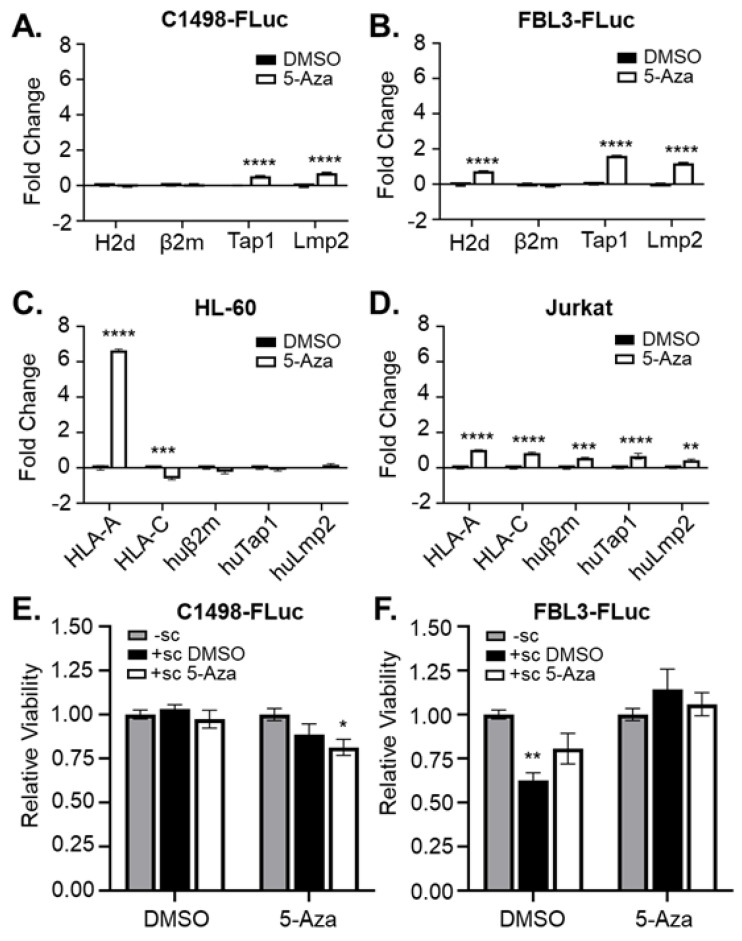
5-Aza treatment rescues expression of genes related to antigen presentation. (**A**) C1498-Fluc, (**B**) FBL3-FLuc, (**C**) HL-60, and (**D**) Jurkat cells were treated for 48 h in culture with vehicle (DMSO) and 1 μM 5-Aza (**A**,**B**) or 5 μM 5-Aza (**C**,**D**). RNA was isolated and amplified by qPCR with primers for H2d, β2m, Tap1, and Lmp2 (**A**,**B**) or HLA-A, HLA-C, huβ2m, huTap1, huLmp2 (**C**,**D**). *n* = 3 technical replicates per group. ** *p* < 0.01, *** *p* < 0.001, **** *p* < 0.0001, 2-way ANOVA followed by Sidak’s test for multiple comparisons. (**E**) C1498-Fluc and (**F**) FBL3-Fluc cells were treated in culture with vehicle (DMSO) or 5 μM 5-Aza for 24 h before washing and co-culture without splenocytes (-sc) or with splenocytes harvested from mice treated with a single dose of vehicle (+ sc DMSO) or 5mg/kg Aza (+sc 5-Aza) for 24 h. Viability of the culture was measured using MTT and absorbance was read at 590 nm. Viability relative to culture without splenocytes is shown (with viability from splenocytes cultured alone subtracted). * *p* < 0.05, ** *p* < 0.01, unpaired *t*-test.

**Figure 4 cancers-15-00118-f004:**
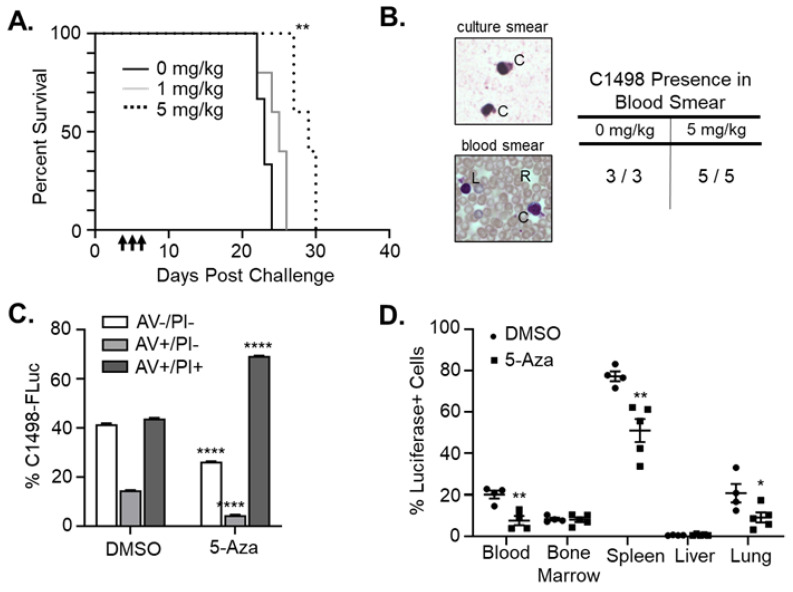
5-Aza decreases leukemic burden and extends survival in immune-competent mice. (**A**) C57Bl/6 mice were challenged with 8 × 10^5^ C1498-FLuc cells via tail vein and treated with vehicle (0mg/kg, *n* = 3), 1mg/kg 5-Aza (*n* = 5 mice), or 5 mg/kg 5-Aza (*n* = 5 mice) for three consecutive days beginning on day 4 post challenge. Percent survival by days post challenge is shown as a Kaplan–Meier curve. ** *p* < 0.01, Log-rank (Mantel-Cox) test and Gehan-Breslow-Wilcoxon test. (**B**) Retro-orbital blood collection on mice from (**A**) was performed on Day 17 post-challenge. Blood was smeared on a microscope slide and Wright’s stain was used to differentiate red blood cells (R) from white blood cells including engrafted C1498-Fluc (C) and lymphocytes (L). Representative images of C1498-Fluc from in vitro culture and in the blood of engrafted mice are shown. All mice from 0 mg/kg and 5 mg/kg 5-Aza treatment groups were confirmed to have C1498 present in blood smears (table). (**C**) C1498-FLuc cells in vitro were treated with vehicle (DMSO) or 5-Aza (5 μM) for 48 h and stained with antibody recognizing Annexin V (AV) and propidium iodide (PI). Percentage out of total cells of viable (AV−/PI−), early apoptotic (AV+/PI−), and late apoptotic (AV+/PI+) cells are plotted as a bar graph. Necrotic cells (AV−/PI+) cells were negligible in all samples. *n* = 3 wells per group. **** *p* < 0.0001, two-way ANOVA followed by Sidak’s multiple comparisons test. (**D**) C1498-FLuc cells were implanted in C57Bl/6 mice as in (**A**) and treated with vehicle (DMSO) or 5-Aza (5 mg/kg). Forty-eight hours after the third treatment, blood, bone marrow, spleen, liver, and lung from mice were reduced to single cell suspension and stained intracellularly with antibody recognizing luciferase followed by anti-rabbit IgG fluorescent secondary antibody. Percentages of luciferase positive cells out of total cells were analyzed by flow cytometry and are presented as a bar graph. * *p* < 0.05, ** *p* < 0.01 unpaired *t*-test with Welch’s correction.

**Figure 5 cancers-15-00118-f005:**
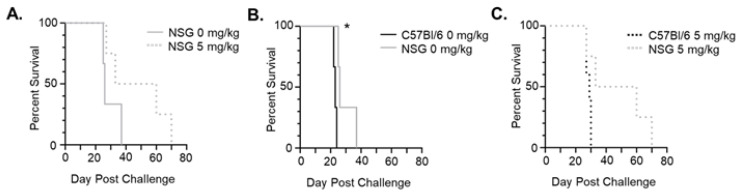
5-Aza extends survival in half of immuno-deficient mice. (**A**) NSG mice were challenged with 8 × 10^5^ C1498-FLuc cells via tail vein and treated with vehicle (0 mg/kg) (*n* = 3 mice) or 5 mg/kg 5-Aza (*n* = 4) for three consecutive days beginning on day 4 post-challenge. Percent survival by days post-challenge is shown as a Kaplan–Meier curve. (**B**) Comparison of survival between vehicle (0mg/kg) treated C57Bl/6 and NSG mice challenged with C1498-Fluc. (**C**) Comparison of survival between 5-Aza-treated (5 mg/kg) C57Bl/6 and NSG mice challenged with C1498-Fluc. * *p* < 0.05, Log-rank (Mantel-Cox) test and Gehan-Breslow-Wilcoxon test.

**Figure 6 cancers-15-00118-f006:**
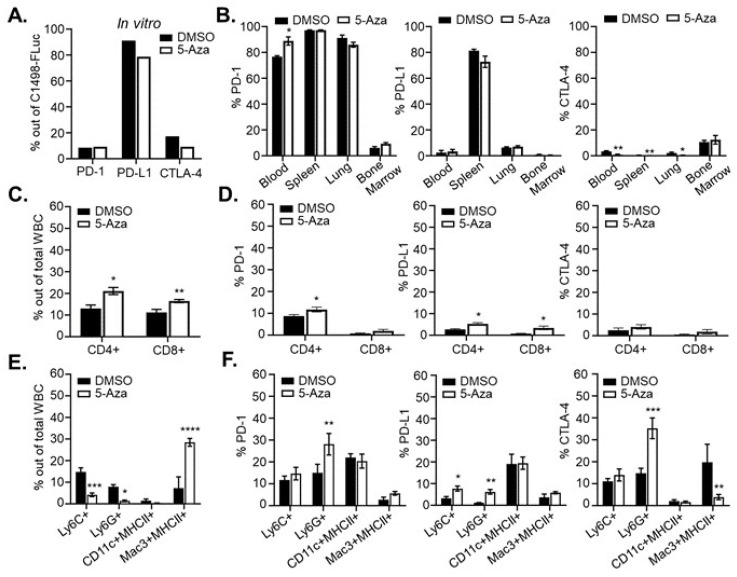
5-Aza treatment alters the immune microenvironment. (**A**) Two million C1498-Fluc cells were treated with 10 µM 5-Aza for 48 h in culture and stained with antibodies recognizing PD-1, PD-L1, and CTLA-4. Percentages of positive cells analyzed by flow cytometry are shown for vehicle (DMSO) and 5-Aza treated cells. (**B**) C57Bl/6 mice were challenged with 8 × 10^5^ C1498-FLuc cells via tail vein and AML was allowed to develop. On day 14 post-challenge, mice were given three consecutive, daily treatments of vehicle (DMSO) (*n* = 4 mice) or 5 mg/kg 5-Aza (5-Aza) (*n* = 5 mice). Forty-eight hours after the third treatment, blood, spleen, lung, and bone marrow was processed for flow cytometry. Cells were surface stained with antibodies recognizing PD-1 and PD-L1 as well as stained intracellularly for CTLA-4 and Luciferase. Percentages of PD-1, PD-L1 or CTAl-4 positive cells gated out of Luciferase positive cells analyzed by flow cytometry are shown. (**C**–**F**) Mice were treated as in (**B**) and processed blood samples were stained with antibodies recognizing CD4, CD8, PD-1, PD-L1, and CTLA-4 (**C**,**D**). One hundred thousand cells (±1%) were analyzed by flow cytometry and percentages of positive cells are shown out of total WBC or gated CD8+ or CD4+ positive cells. (**E**,**F**) Processed blood cells were stained with antibodies recognizing Ly6C, Ly6G, CD11c, MHCII, Mac3 (intracellular), PD-1, PD-L1, and CTLA-4 (intracellular). Percentages of positive cells analyzed by flow cytometry are shown out of total WBC or gated myeloid populations. For all panels (**A**–**F**) * *p* < 0.05, ** *p* < 0.01, *** *p* < 0.001, **** *p* < 0.0001, unpaired *t*-test with Welch’s correction.

## Data Availability

No new data were created or analyzed in this study. Data sharing was not applicate to this article.

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
