# Peer review of "5-Azacytidine-Mediated Modulation of the Immune Microenvironment in Murine Acute Myeloid Leukemia"

_cancers, 2022, doi:10.3390/cancers15010118_

Round 1

Reviewer 1 Report (Previous Reviewer 3)

The Authors have improved the Manuscript to some extent. However, my main criticisms were only partially addressed. 

Author Response

The Authors have improved the Manuscript to some extent. However, my main criticisms were only partially addressed.

We thank the Reviewer for again taking the time to thoroughly critique our work. As the Reviewer has not clarified which criticisms remain, we have included below their original comments and our responses, as well as additional responses for this round of revisions in italics. We have focused on further revising our interpretation of data from the NSG experiment and have removed references to "pro-leukemic" cells (as interpreted from 5B). We hope the Reviewer will find our interpretations of the data to be more conservative.

  1. Overall, despite the title, all experiments were performed with cell lines (murine mostly) and neither primary AML samples (ex-vivo and in PDX model in vivo) nor immune cells were studied. Therefore it is hard to draw any firm conclusion on the interplay between AML cells, AZA and the immune microenviroment when only leukemic cell lines were tested.

We have updated the title of the manuscript to highlight that these studies occur in pre-clinical murine models of AML (page 1, line 3). With our revised manuscript, we hope to support the use of the C1498 and FBL3 models in deciphering the mechanisms of interplay between 5-Aza treatment, leukemic cells, and the immune microenvironment in human leukemia as well as begin to identify markers of resistance, things not possible using an immune-deficient (xenograft) model. Studies using primary AML samples have elucidated genetic signatures for response to hypomethylating agents (PMID 28278722), and a pre-clinical model may help identify markers at other levels of gene expression (e.g. protein expression, processing, localization) as well as clarify the mechanisms of how gene expression and immune subset changes lead to response or resistance. We have more thoroughly explored these points in the introduction (page 2, lines 66-82).

  1. Results section 189-192. It is not clear to me why the lack of difference in survival curves between mice implanted with parental C1498 and C1498-FLuc may be due to immune-suppression. Is there any prior data suggesting that tumor transduced cells induce different survival as compared to the parental ones ? What are the data demonstrating the activity of immune system ?

We thank the reviewer for pointing out this deficiency. Studies in multiple tumor models have reported differing growth and survival curves between parental tumor cell lines and FLuc transduced cell lines, as well as specific immune activity against FLuc antigens. We have cited and discussed these studies in the Results section 3.1 (page 5, lines 209-224).

  1. Results section 213-217 AZA treatment induces re-expression of the transgene through epigenetic silencing. However, there no data to support the conclusion that AZA increases the immunogenicity of AML cells.

Based on the reviewer’s comment, we have now included new functional immunogenicity data showing that Aza treatment of C1498 cells increases their susceptibility to killing by splenocytes (harvested from DMSO or Aza-treated mice), indicating an increase in immunogenicity of AML cells (Figure 3E&F, page 7 lines 290-300). No killing by splenocytes is observed when using untreated AML cells.

  1. Results section 275 279. The modulation of antigen-presentation machinery by AZA is well taken. However, without functional data on effector cells and APC it is hard to translate this information into the conclusion that activation of the immune system is able to prolong survival of AZA_treated mice.

As mentioned above, new data from a functional immunity assay have been added as Figure 3E and 3F and are discussed in the Results section 3.3 (page 7 lines 290-300).

  1. I have problems wih Fig.5. First of all, the statistics of A, B and C are not totally convincing. In A 50% survival is approximately 25 days vs not reached at 50 days; in C 30 days vs not reached at 50 days. Both comparisons were not statistically significant. However, in B the difference between 24 days (?) and 26 days (?) was significant. In addition, even if we assume that data analysis is correct it is not clear to me which is the role of the immune system in AZA-treated mice. Indeed, based on background experiments one would have expected an inferior survival curve in NSG mice. The Authors claim the role of pro-leukemic immune cells in immune competent mice (Tregs ?). However, no data are provided in this regard.

We thank the reviewer for these comments. Statistical analysis was done using Prism 9 from Graphpad. As we agree that while the analysis did not show significance, there are clear responses to 5-Aza treatment in the NSG mice, hence, we have tried to interpret the data with this in mind. We have, however, updated our interpretation of these results to reflect that these conclusions are not fully supported at this time and require additional future experiments to support our claims (Results section 3.5 page 10 lines 380-397; Discussion pages 13-14 lines 533-544). Although the difference in 5A appears larger than in 5B, statistically only the difference between curves in 5B is significant, possibly due to the tightness of the curves compared to those in 5A and 5C. p-values have been added in the text for each figure panel for clarification (Page 10, Lines 367-368, and 381-384). Despite lack of statistical significance, however, we do discuss the results based on obvious increased median survival in the 5-Aza treated group. Furthermore, we have revised both the Results section 3.5 as well as the Discussion to no longer include the supposition of pro-leukemic cells based on the interpretation of the NSG vs Bl6 data. We have also included that the small difference in survival curves between vehicle treated NSG and Bl6 mice may be due to differences in the model beyond the immune competence (Pages 10, lines 377-389).

  1. Results 373-377. The expression of PD-1 on leukemic cell lines should be discussed as well CTLA-4. The Authors state that AZA mai first induce activation of T cells followed by dampened immune response shortly after. Again, there are no data to support such conclusion.

We have further elaborated on the expression of the checkpoint molecules PD-1, PD-L1 and CTLA-4 (Results section 3.6), including references describing the expression of these proteins in human AML. We have updated the discussion to include theories on how the pro-leukemic immune subsets may be MDSC and T-cells with increased expression of immune checkpoint proteins (supported by our flow cytometric analyses). We have also mentioned that further mechanistic studies, including extensive antibody-mediated, immune subset depletions, will be needed to confirm these hypotheses (Discussion pages 13-14, lines 538-540). We have added a graphic summarizing the effects of 5-Aza on PD-1, CTLA-4 and PD-L1 expression on leukemic cells as well as immune cells in the surrounding microenvironment (Graphical Abstract) based on comments from another Reviewer.

Reviewer 2 Report (Previous Reviewer 2)

Major

1. The survival difference in 5B was modest, difference in 5A seems to be larger than 5B; what was the p-value in 5A? The host mice number N wasn’t updated in the legend. This result will affect the conclusion/title in section 3.5.

2. 5B, under the condition without Aza treatment, if the transplantation experiments were set up exactly the same (cell dose, infusion route, etc), inferior survival is expected in the NSG group. Result in 5B may be due to the specific growth kinetics of this model in NSG, or many other factors may also play a role. For instance, it has been shown that leukemic cell grows better in female NSG recipients thus the mice succumb to death faster. What I am trying to express is the complexity the of the mouse models, and it’s very hard to draw any solid conclusion regarding the role of immune system in Aza treatment response by comparing B6 and NSG. The authors showed effects of Aza treatment in immune system in Figure 6 with B6, I would focus on that in the manuscript if no additional data is provided with the NSG model.

Author Response

Thank you again for your thorough review of our work, and we appreciate the further clarification of your original statements. We feel we were more able to resolve your concerns with this round of revisions. Please find our responses below. 

  1. The survival difference in 5B was modest, difference in 5A seems to be larger than 5B; what was the p-value in 5A? The host mice number N wasn’t updated in the legend. This result will affect the conclusion/title in section 3.5.

Although the difference in 5A appears larger than in 5B, statistically only the difference between curves in 5B is significant, possibly due to the tightness of the curves compared to those in 5A and 5C. p-values have been added in the text for each figure panel for clarification (Page 10, Lines 367-368, and 381-384). Despite lack of statistical significance, however, we do discuss the results based on obvious increased median survival in the 5-Aza treated group.

The n number of mice for these experiments is unchanged as 5-Aza mice that had not succumbed to disease by the time of the initial submission were merely censored from the Kaplan-Meier (a value of “0” entered), but were still present in the analysis. The title of section 3.5 has been updated to reflect that a subset of engrafted NSG mice do respond to 5-Aza treatment (Page 10, Lines 360-361).

  1. 5B, under the condition without Aza treatment, if the transplantation experiments were set up exactly the same (cell dose, infusion route, etc), inferior survival is expected in the NSG group. Result in 5B may be due to the specific growth kinetics of this model in NSG, or many other factors may also play a role. For instance, it has been shown that leukemic cell grows better in female NSG recipients thus the mice succumb to death faster. What I am trying to express is the complexity the of the mouse models, and it’s very hard to draw any solid conclusion regarding the role of immune system in Aza treatment response by comparing B6 and NSG. The authors showed effects of Aza treatment in immune system in Figure 6 with B6, I would focus on that in the manuscript if no additional data is provided with the NSG model.

This point is well taken and we appreciate the reviewer’s further clarification on this topic. We have revised both the Results section 3.5 as well as the Discussion to no longer include the supposition of pro-leukemic cells based on the interpretation of the NSG vs Bl6 data. We have also included that the small difference in survival curves between vehicle treated NSG and Bl6 mice may be due to differences in the model beyond the immune competence (Pages 10, lines 377-389).

Reviewer 3 Report (New Reviewer)

This is a very interesting article and findings on Acute Myeloid Leukemia.

In this article, the authors investigated that 5-Aza therapy restored immune-related transcript expression, reduced the burden of leukemia, and increased survival in mice with leukemia. The authors also showed that the effects of 5-Aza therapy were transient, and examination of the immunological milieu shows potential resistance mechanisms, such as concurrent elevation of immune checkpoint protein expression. The further author stated that this is a model system that exhibits great sensitivity to HMAs, recapitulates key therapeutic outcomes seen in human leukemia (relapse), and may be used as a pre-clinical tool to investigate acquired resistance and innovative treatment approaches. The paper is well organized and written, however, there are a few suggestions:

Minor concerns:

  1. The authors should increase the quality/size of figure 2 for better visualization.
  2. The authors should include detailed methodology in the method section.

Author Response

We thank the Reviewer for their thorough review. Please find our responses to your concerns below.

  1. The authors should increase the quality/size of figure 2 for better visualization.

We thank the reviewer for these suggestions. We have increased the size of Figure 2 A and B as well as included tiff formats (separate files) that are higher quality than those in the word document (Page 10, Line 251).

  1. The authors should include detailed methodology in the method section.

We have added details to most sections of the Materials and Methods. We hope the reviewer will find these included details to be satisfactory. (Pages 3-4, Lines 106, 126, 132, 140-141, 145-146, 158-169, 187-188, and 195)

Reviewer 4 Report (New Reviewer)

 The present manuscript authored by Nancy D. Ebelt and Edwin R. Manuel entitled “5-Azacytidine-Mediated Modulation of the Immune Microenvironment in Acute Myeloid Leukemia is up for consideration to publish in MDPI-Cancers journal.

Overall, the manuscript is straightforward, well written, concise, and clear within the scope of MDPI-Cancers.

 However, I have minor queries which should be addressed before publishing this article:

(1)   The authors should explain the rationale behind using the cell lines HL60 and Jurkat. Did they perform similar experiments with other leukemic cells?

(2)   Line 66: correlated to a

(3)   Line70: regards

(4)   Line98: decreased

(5)   Line237: regards

(6)   A final figure/cartoon showing the gist of the study would be very helpful.

Author Response

  1. The authors should explain the rationale behind using the cell lines HL60 and Jurkat. Did they perform similar experiments with other leukemic cells?

We thank the reviewer for their detailed observations. These experiments were only performed with HL60 and Jurkat cell lines, hence they were the only human cell line data included in Figure 3. We have elaborated on their typing and rationale for their use in Results section 3.3 (Page 7, Lines 278 and 287-289) with additional supporting citations.

  1. Line 66: correlated to a,Line70: regards, Line98: decreased, Line237: regards

We thank the reviewer for their suggestions, these edits have now been updated in the text.

  1. A final figure/cartoon showing the gist of the study would be very helpful.

The Cancers journal recommends the use of graphical abstracts in addition to the written abstract, thus we have included a summary illustration as a graphical abstract (see uploaded Figure PDF). Thank you for this suggestion.

Round 2

Reviewer 1 Report (Previous Reviewer 3)

The Authors have certainly improved the manuscript and data interpretation. However, the translational potential of their work remains low: without a AML model built on human cell lines and further supported by primary cells its merit is modest.  

Author Response

We thank the reviewer for their continued evaluation of our manuscript and for their important criticisms. We agree that the impact of the manuscript would be improved by the addition of a human AML model or inclusion of primary samples, however, we have contended in the Introduction that transplantable mouse models of leukemia in immune competent mice have benefits over other options and should be utilized.

On page 2, lines 77-84 we state that “Transplantable mouse models of leukemia have many advantages: 1) the convenience of the rapid onset of disease that allows for immediate testing of experimental therapies, 2) immune competent mouse models support more mechanistic studies of disease progression (e.g. genetic manipulation, specific cell depletion) compared to correlative gene expression analysis in primary human samples, and 3) patient-derived xenograft or human cell line models using humanized mice lack fully functioning immune systems that may not correctly recapitulate leukemic interaction with the immune microenvironment [PMC7518015].”

As we agree with the reviewer that other models would lend confirmation and impact to the conclusions of this study, we highlight cited clinical data in the Discussion that already agrees with our findings (page 13, lines 507-511, lines 517-520, and lines 528-531), as well as add comments that continued confirmation of the pre-clinical mouse model data with primary human samples and more advanced humanized mouse models will be needed to fully dissect the mechanisms of interplay between leukemic cells and the immune microenvironment after treatment with 5-Aza. We hope to be able to add these important experiments to future studies (Pages 13 and 14, Lines 541-543 and 559-561).

Reviewer 2 Report (Previous Reviewer 2)

No further comments

Author Response

We thank the reviewer for taking additional time to thoroughly assess our work. Additional minor revisions including spell check and language editing can be found on page 1, line 22; page 2, line 85; page 3, line 132; page 5, line 203; and page 12, line 452.

This manuscript is a resubmission of an earlier submission. The following is a list of the peer review reports and author responses from that submission.

Round 1

Reviewer 1 Report

Accumulation of epigenetic modifications may help tumour cells escaping from the immune system, and therefore, re- sensitize these tumour cells to the immune system would seem therapeutically very promising.

In this context, the authors have analysed the selective silencing of genes involved in immune detection in leukemic cells, and whether such genes could be re-activated by the treatment with the hypomethylating agent (HMAs), 5-aza-9 cytidine (5-AZA). The authors address this issue in vitro and in vivo. They use syngeneic leukemic models in which they injected a murine leukemic cell line that carries a luciferase reporter. They observed that such reported is rapidly silenced in the mice. 5-AZA treatment re-stablish the expression of the transgene in the leukemic cells, and what is more important, enhances the expression of immune-related transcripts in the injected cells, what increases their immunogenicity by the upregulation of antigen presentation-related transcripts. However, at the same time 5-AZA treatment also enhances immune suppressive pathways by enhancing the expression of PD-1 and PDL-1.

This study is very interesting and highlights the importance of the immune system to eradicate leukemic burden, and strongly suggests that simultaneous treatment with HMAs and immune checkpoint might be an interesting therapeutic strategy. Moreover, the study provides syngeneic leukemic models where to test such possibility in the future.

Reviewer 2 Report

Summary:

Immune checkpoint molecule upregulation is believed to be an important mechanism of 5-Azacytidine (Aza) resistance, and combination of hypomethylating agents (HMA) and immune checkpoint blockage has showed encouraging therapeutic efficacy in preclinical and clinical studies. Understanding the effect and potential mechanism of Aza in immune modulation has significant clinical impact.

In the present study, the authors evaluated the activity of hypomethylating agents 5-Azacytidine (Aza) with a syngenetic murine leukemia model established with C1489-Fluc cells. The authors found that Aza treatment inhibited the growth of C1489 cells in vitro and in vivo, and prolonged the survival of mice infused with C1489-Fluc. Moreover, the authors found that Aza treatment increased the expression of antigen presentation genes in vitro, as well as percentage of PD-1/PD-L1+ T cells in vivo. Mice treated with Aza also showed increased percentage of CD4+ and CD8+ T cells, and decreased Ly6C+ monocytes and Ly6G+ neutrophils. The direct modulation of the immune system maybe one of the underlying mechanisms of Aza’s adaptive resistance.

Comments:

3.1, Figure 1

·       1B, the transplantation model needs to be further validated if the FLuc expression is unstable in vivo. For instance, quantify the FLuc genomic DNA (with qPCR) in blood, bone marrow, and spleen across different time points post cell infusion.

·       1C, and for endpoint analysis, leukemic engraftment needs to be confirmed, especially for in vivo study with low number of host animals.

·       Alternatively, in vivo study can be performed with CD45.2 B6 as host animal, and leukemic engraftment can be assessed by FACS analysis of CD45.1 vs CD45.2 (e.g. PMID: 27768040).

3.3, Figure 3

·       what is Y? And needs to confirm that Aza increases the surface/whole cell level of antigen presenting proteins.

·       Did the authors also observe increase of these genes on C1489 cells from in vivo model?

3.4, Figure 4

·       4A, again, in additional to the survival analysis, tumor burden needs to be confirmed at endpoint analysis.

·       4C, if Aza restores the epigenetic silencing of Fluc gene, % Luciferase+ is not a good indicator of tumor burden.

·       4C, what are the spleen and liver weights in Aza compared to control? C1498 transplantation model showed enlarged spleen and liver according to published studies.

3.5, Figure 5

·       5A-C, number of host mice was too low to draw a solid conclusion, at least double the number.

·       5A, three consecutive 5mg/kg is a relatively high Aza dose, most of the NSG leukemic models are expected to show good response to this dose. The results shown may be due to the low number of host mice.

·       This dose can also be ‘toxic’ for NSG mice. Was irradiation performed before cell infusion? If yes, host mice may die of Aza treatment mostly because of bone marrow failure. What were the leukemic engraftments in both groups? Was low cellularity observed?

·       5B-C, the growth kinetics of C1489 cells can be totally different in B6 and NSG due to the huge differences in the niche, such comparison of survival is not informative; To show the contribution of immune system, such assay should be performed with NSG mice, with and without co-transplantation with murine PBMC or BM cells (e.g. from CD45.1 models).

3.6, Figure 6

·       6B-F, are these measurements from mice showed response or resistance to Aza treatment?

·       6C-D, the authors reported increase percentage of CD4+ and CD8+ cells upon Aza treatment, this may also be due to the decrease of other cell lineages (for instance, 6E-F); did the absolute numbers of CD4+ or CD8+ T cells also increase upon Aza as observed in PMID: 23242597?

Reviewer 3 Report

The manuscript by Ebelt and Manuel addresses the AZA-mediated modulation of the immune microenviroment in AML.

The paper suffers of major problems which should be addressed by the Authors.

Major:

a) Overall,  despite the title, all experiments were performed with cell lines (murine mostly) and neither  primary AML samples (ex-vivo and in PDX model in vivo) nor immune cells were studied. Therefore it is hard to draw any firm conclusion on the interplay between AML cells, AZA and the immune microenviroment when only leukemic cell lines  were tested.

b)Results section 189-192. It is not clear to me why the lack of difference in survival curves between mice implanted with parental C1498 and C1498-FLuc may be due to immunesuppression. Is there any prior data suggesting that tumor transduced cells induce different survival as compared to the parental ones ? What are the data demonstrating the activity of immune system ? 

c) Results section 213-217 AZA treatment induces re-expression of the transgene through epigenetic silencing. However, there no data to support the conclusion that AZA increases the immunogenicity of AML cells. 

d) Results section 275 279. The modulation of antigen-presentation machinery by AZA is well taken. However, without functional data on effector cells and APC it is hard to translate this information into the conclusion that activation of the immune system is able to prolong survival of AZA_treated mice.

e) I have problems wih Fig.5. First of all, the statistics of A, B and C are not totally convincing. In A 50% survival is approximately 25 days vs not reached at 50 days; in C 30 days vs not reached at 50 days. Both comparisons were not statistically significant. However, in B the difference between 24 days (?) and 26 days (?) was significant. In addition, even if we assume that data analysis is correct it is not clear to me which is the role of the immune system in AZA-treated mice. Indeed, based on background experiments one would have expected an inferior survival curve in NSG mice. The Authors claim the role of pro-leukemic immune cells in immune competent mice (Tregs ?). However, no data are provided in this regard.

f) Results 373-377. The expression of PD-1 on leukemic cell lines should be discussed as well CTLA-4. The Authors state that AZA mai first induce activation of T cells followed by dampened immune response shortly after. Again, there are no data to support such conclusion.